# Rare variants contribute disproportionately to quantitative trait variation in yeast

Joshua S Bloom[1,2,3,4]*, James Boocock[1,2,3,4], Sebastian Treusch[1,2,3,4†], Meru J Sadhu[1,2,3,4‡], Laura Day[1,2,3], Holly Oates-Barker[1,2,3], Leonid Kruglyak[1,2,3,4]*

[1]Department of Human Genetics, University of California, Los Angeles, Los Angeles, United States; [2]Department of Biological Chemistry, University of California, Los Angeles, Los Angeles, United States; [3]Howard Hughes Medical Institute, University of California, Los Angeles, Los Angeles, United States; [4]Institute for Quantitative and Computational Biology, University of California, Los Angeles, Los Angeles, United States

**Abstract** How variants with different frequencies contribute to trait variation is a central question in genetics. We use a unique model system to disentangle the contributions of common and rare variants to quantitative traits. We generated ~14,000 progeny from crosses among 16 diverse yeast strains and identified thousands of quantitative trait loci (QTLs) for 38 traits. We combined our results with sequencing data for 1011 yeast isolates to show that rare variants make a disproportionate contribution to trait variation. Evolutionary analyses revealed that this contribution is driven by rare variants that arose recently, and that negative selection has shaped the relationship between variant frequency and effect size. We leveraged the structure of the crosses to resolve hundreds of QTLs to single genes. These results refine our understanding of trait variation at the population level and suggest that studies of rare variants are a fertile ground for discovery of genetic effects.

DOI: https://doi.org/10.7554/eLife.49212.001

*For correspondence:
jbloom@mednet.ucla.edu (JSB);
LKruglyak@mednet.ucla.edu (LK)

Present address: †Intrexon, South San Francisco, United States; ‡National Human Genome Research Institute, National Institutes of Health, Bethesda, United States

## Introduction

A detailed understanding of the sources of heritable variation is a central goal of modern genetics. Genome-wide association studies (GWAS) in humans (*Visscher et al., 2017*) have implicated tens of thousands of DNA sequence variants in disease risk and quantitative trait variation, but these variants fail to account for the entire heritability of diseases and traits. One key question is the relative contribution of DNA sequence variants with different allele frequencies in a population to trait variation. GWAS by design only test common DNA sequence variants; however, recent studies underscore the likely importance of the contribution of rare variants to heritable variation (*Wainschtein et al., 2019*). Theoretical analyses have explored how factors such as mutational target size, pleiotropy, and the strength of selection shape the relationship between variant frequency and effect size (*Eyre-Walker, 2010*; *Robinson et al., 2014*; *Simons et al., 2018*). In particular, purifying selection against variants that negatively affect fitness is expected to keep them at low frequencies in a population, resulting in a predicted inverse relationship between effect sizes and allele frequencies for variants that influence fitness-related traits (*Gibson, 2012*; *Goldstein et al., 2013*; *Kryukov et al., 2007*; *Pritchard, 2001*).

Empirical results have been consistent with the theoretical expectation that rare variants should have larger effect sizes, or, equivalently, that variants implicated in trait variation should be shifted to lower frequencies relative to all variants. An increased burden of ultra-rare protein-truncating

variants has been observed in human diseases (**Ganna et al., 2018**; **Exome Aggregation Consortium et al., 2016**), and multiple studies have found that GWAS variants with lower allele frequencies have larger effect sizes (**Marouli et al., 2017**; **Park et al., 2011**). A negative correlation between allele frequency and effect size has also been observed in maize GWAS (**Wallace et al., 2014**), and our previous work in yeast suggested that variants that contribute to trait variation are shifted to lower frequencies when compared to all sequence variants (**Ehrenreich et al., 2012**).

Recent studies employed indirect variance partitioning approaches to uncover appreciable contributions of lower frequency variants to heritability of complex traits in humans, including prostate cancer susceptibility (**Mancuso et al., 2016**), height (**Wainschtein et al., 2019**; **Yang et al., 2015**), and body mass index (**Wainschtein et al., 2019**). However, a direct comprehensive comparison of the effects of rare and common variants has been lacking in humans for two principal reasons. First, rare variants cannot be detected by GWAS by design, and sequencing studies have not reached sufficient sample sizes to find them with high statistical power (**Zuk et al., 2014**). As a result, most rare variants have to date escaped detection. Second, the power to detect a variant with any given effect size decreases with the frequency of the variant in the study, simply because fewer individuals in the sample carry a less-frequent variant (**Zuk et al., 2014**). This statistical artifact shifts the effect sizes of those rare variants that are detected upwards, confounding effect size and allele frequency and biasing any effort to measure the underlying relationship between the two.

Here, we report a comprehensive study in yeast designed to overcome these limitations. We built a mapping population consisting of approximately one thousand progeny from each of 16 biparental crosses. In this mapping population, even variants that are rare in the yeast population and occur in only a single parental strain are present in approximately 1000 progeny, resulting in high power to detect them. We mapped thousands of QTLs that account for most of the heritable variation in 38 quantitative traits and measured the QTL effect sizes. We then decoupled variant frequency from effect size by measuring the population allele frequencies of QTL lead variants detected in our panel in a separate large catalog of sequenced yeast isolates (**Peter et al., 2018**). Analysis of these large complementary data sets enabled us to directly and comprehensively examine the relationship between QTL effect sizes and variant frequency, characterize the genetic architecture of quantitative traits on a population scale, and improve mapping resolution, in many cases to single genes.

## Results

To investigate the genetic basis of quantitative traits in the yeast population, we selected 16 highly diverse *S. cerevisiae* strains that capture much of the known genetic diversity of this species. Specifically, they contain both alleles at 82% of biallelic SNPs and small indels observed at minor allele frequency >5% in a collection of 1011 *S. cerevisiae* strains (**Peter et al., 2018**). We sequenced the 16 strains to high coverage in order to obtain a comprehensive set of genetic variants. We constructed a panel of 13,950 individual recombinant haploid yeast segregants by crossing each parental strain to two different strains and collecting an average of 872 progeny per cross (**Figure 1**; **Figure 1—source data 1**; **Supplementary file 1**). We genotyped these segregants by highly multiplexed whole-genome sequencing, with median 2.3-fold coverage per base per individual. Genotypes were called at 298,979 genetic variants, with an average of 71,117 genetic variants segregating in a single cross. We measured the growth of each segregant in 38 different environments in duplicate by automated assays and quantitative imaging (Materials and methods). Because the growth measurements in different environments are not strongly correlated, we treat them as separate phenotypes or traits (**Bloom et al., 2013**). The resulting genotype-by-phenotype matrix (over half a million phenotypic measurements and 158 billion combinations of genotype and phenotype) formed the basis for all downstream analyses.

We used a variance components model (**Bloom et al., 2015**; **de los Campos et al., 2015**; **Yang et al., 2010**) to show that, on average, additive genetic effects accounted for just over half of the total phenotypic variance, while pairwise genetic interactions accounted for 8%, approximately 1/6 as much as additive effects (**Figure 2** inset; **Supplementary file 2**; **Figure 2—source data 1**). We carried out QTL mapping to find the specific loci contributing additively to trait variation. We used a joint mapping approach that leverages information across the entire panel of 13,950 segregants (Materials and methods). We mapped 4552 QTLs at a false discovery rate (FDR) of 5%, with an average of 120 (range 52–195) QTLs per trait (**Supplementary file 3**; **Figure 3—source data 1**). The

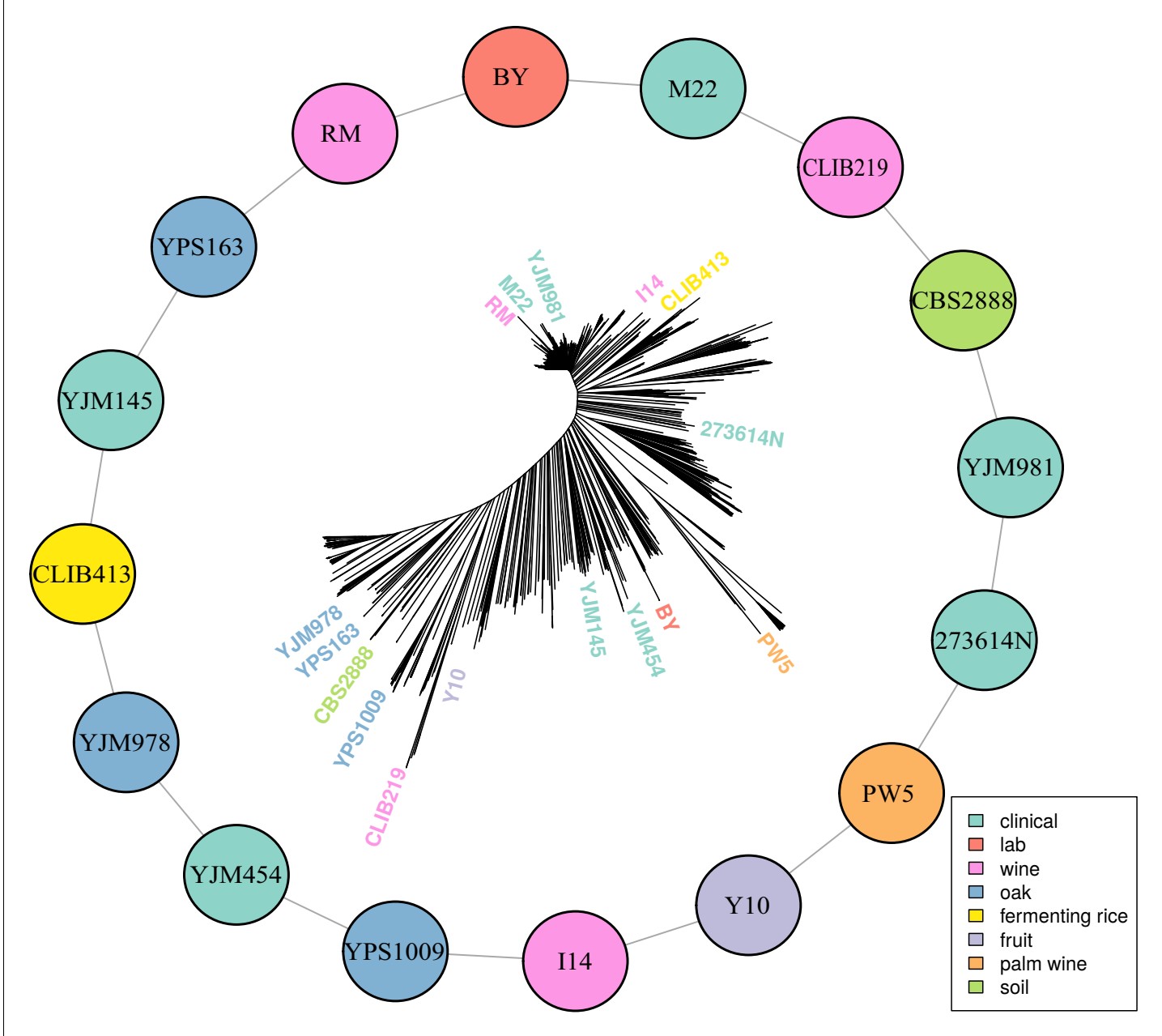

**Figure 1.** Multiparental cross design with 16 diverse progenitor yeast strains. 16 parental strains were chosen to represent the diversity of the *S. cerevisiae* population, as illustrated by their positions on a neighbor-joining tree based on 1011 sequenced isolates (*Peter et al., 2018*). These strains were crossed in a single round-robin design, with each strain crossed to two other strains, as depicted by lines connecting the colored circles. Colors indicate the ecological origins of the parental strains.

DOI: https://doi.org/10.7554/eLife.49212.002

The following source data is available for figure 1:

**Source data 1.** Additional information on yeast crosses and phenotypes.

DOI: https://doi.org/10.7554/eLife.49212.003

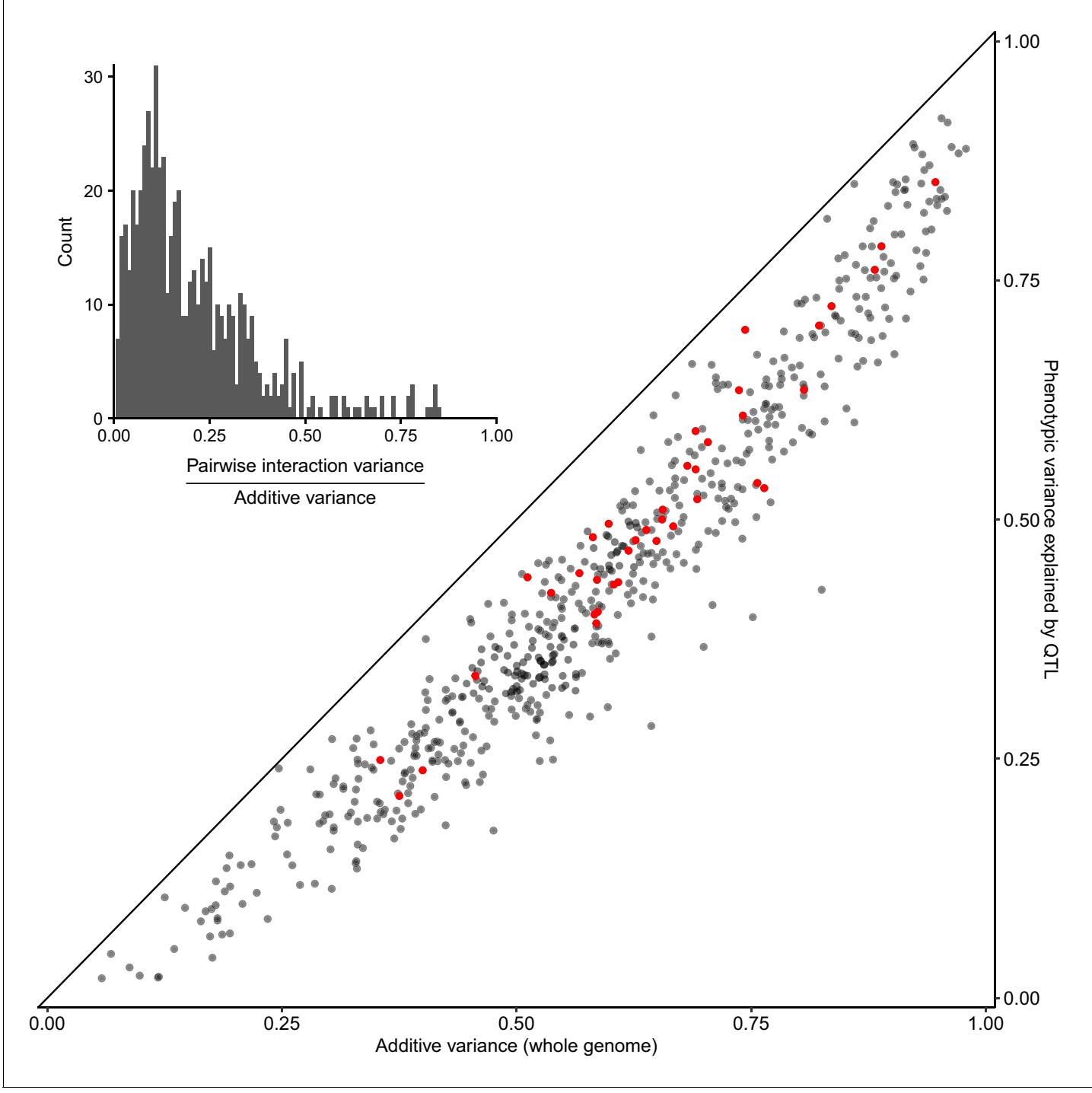

**Figure 2.** Most heritable variation is explained by detected QTLs. Whole-genome estimates of additive genetic variance (X-axis) are plotted against cross-validated estimates of trait variance explained by detected QTLs (Y-axis) for each trait-cross combination. Red points show values for the BY-RM cross. The diagonal line corresponds to detected QTLs explaining all of the estimated additive genetic variance, and is shown as a visual guide. (Inset) A histogram of the ratio of non-additive to additive genetic variance for each trait-cross combination, based on estimates from a variance component model.

DOI: https://doi.org/10.7554/eLife.49212.004

The following source data is available for figure 2:

**Source data 1.** Total variance explained by QTLs and within-cross variance component analyses.

DOI: https://doi.org/10.7554/eLife.49212.005

detected QTLs explain a median of 73% of the additive heritability per trait and cross, showing that we can account for most of the genetic contribution to trait variation with specific loci (*Figure 2*; *Figure 2—source data 1*). We complemented the joint analysis with QTL mapping within each cross and found a median of 12 QTLs per trait at the same FDR of 5%. The detected loci explained a median of 68% of the additive heritability (*Figure 2—source data 1*). The joint analysis was more powerful, explaining an additional 5% of trait variance and uncovering 458 QTLs not detected within individual crosses. Consistent with the higher statistical power of the joint analysis, these additional QTLs had smaller effect sizes (median of 0.071 SD units vs 0.083 SD units; Wilcoxon rank sum test W = 1e6, p=9e-5). All subsequent results are based on the QTLs detected in the joint analysis.

To investigate the relationship between variant frequency and QTL effects, we focused on biallelic variants observed in our panel whose frequency could be measured in a large collection of 1011 sequenced yeast strains. Based on their minor allele frequency (MAF) in this collection, we designated variants as rare (MAF <0.01) or common (MAF >0.01). By this definition, 27.8% of biallelic variants in our study were rare. For each trait, we computed the relative fraction of variance explained by these two categories of variants in the segregant panel (Materials and methods) (*Yang et al., 2015*). Across all traits, the median contribution of rare variants was 51.7%, despite the fact that they constituted only 27.8% of all variants and that a rare variant is expected to explain less variance than a common one with the same allelic effect size. These results are consistent with rare variants having larger effect sizes and making a disproportionate contribution to trait variation. Comparing different traits, we saw a wide range of the relative contribution of rare variants, from almost none for growth in the presence of copper sulfate and lithium chloride to over 75% for growth in the presence of cadmium chloride, in low pH, at high temperature, and on minimal medium (*Figure 3A*; *Figure 3—figure supplement 1*; *Figure 3—source data 2*). The results for copper sulfate and lithium chloride are consistent with GWAS for these traits in the 1011 sequenced yeast strains—these two traits had the most phenotypic variance explained by detected GWAS loci, which inherently correspond to common variants, with large contributions coming from known common copy-number variation at the *CUP* and *ENA* loci, respectively (*Peter et al., 2018*).

In a complementary analysis, we investigated the relationship between the allele frequency of the lead variant at each QTL and the corresponding QTL effect size. Although the lead variant is not necessarily causal, in our study it is likely to be of similar frequency as the causal variant, and a simulation analysis showed that this approach largely preserves the relationship between frequency and effect size (*Figure 3—figure supplement 2*). Most QTLs had small effects (64% of QTLs had effects less than 0.1 SD units) and most lead variants were common (78%), consistent with previous linkage and association studies. We observed that QTLs with large effects were highly enriched for rare variants, and conversely, that rare variants were highly enriched for large effect sizes (*Figure 3B*; *Figure 3—figure supplement 3*; *Figure 3—figure supplement 4*). For instance, among QTLs with an absolute effect of at least 0.3 SD units, 145 of the corresponding lead variants were rare and only 90 were common. Rare variants were 6.7 times more likely to have an effect greater than 0.3 SD (*Figure 3—source data 1*, Fisher's exact test, p<2e-16). Theoretical population genetics models show that for traits under negative selection, variant effect size is expected to be a decreasing function of minor allele frequency (*Eyre-Walker, 2010*; *Pritchard, 2001*). We empirically observe this relationship in our data for most of the traits examined, providing evidence that they have evolved under negative selection in the yeast population (*Figure 3—figure supplement 5*).

The existence of a close sister species of *S. cerevisiae*—*S. paradoxus*—allowed us to distinguish rare variants by their ancestral state. Variants that share the major allele with *S. paradoxus* are more likely to have arisen in the *S. cerevisiae* population recently than those that share the minor allele with *S. paradoxus*. We classified low-frequency variants as recent or ancient according to whether their major or minor allele was shared with *S. paradoxus*, respectively. Recently arising deleterious alleles have had less time to be purged by negative selection, and therefore recent variants are expected to have stronger effects on gene function, and hence manifest as QTLs with larger effects. Consistent with this expectation, we observed that recent variants were 1.8 times more likely than ancient variants to have an effect size greater than 0.1 SD units (Fisher's exact test p=9e-5) (*Figure 3C*). We further examined the direction of QTL effects and found that recent variants were 1.5 times more likely to decrease fitness (Fisher's exact test p=8e-3). Strikingly, no ancient variant decreased fitness by more than 0.5 SD units, whereas 41 recent variants did (Fisher's exact test p=7e-3).

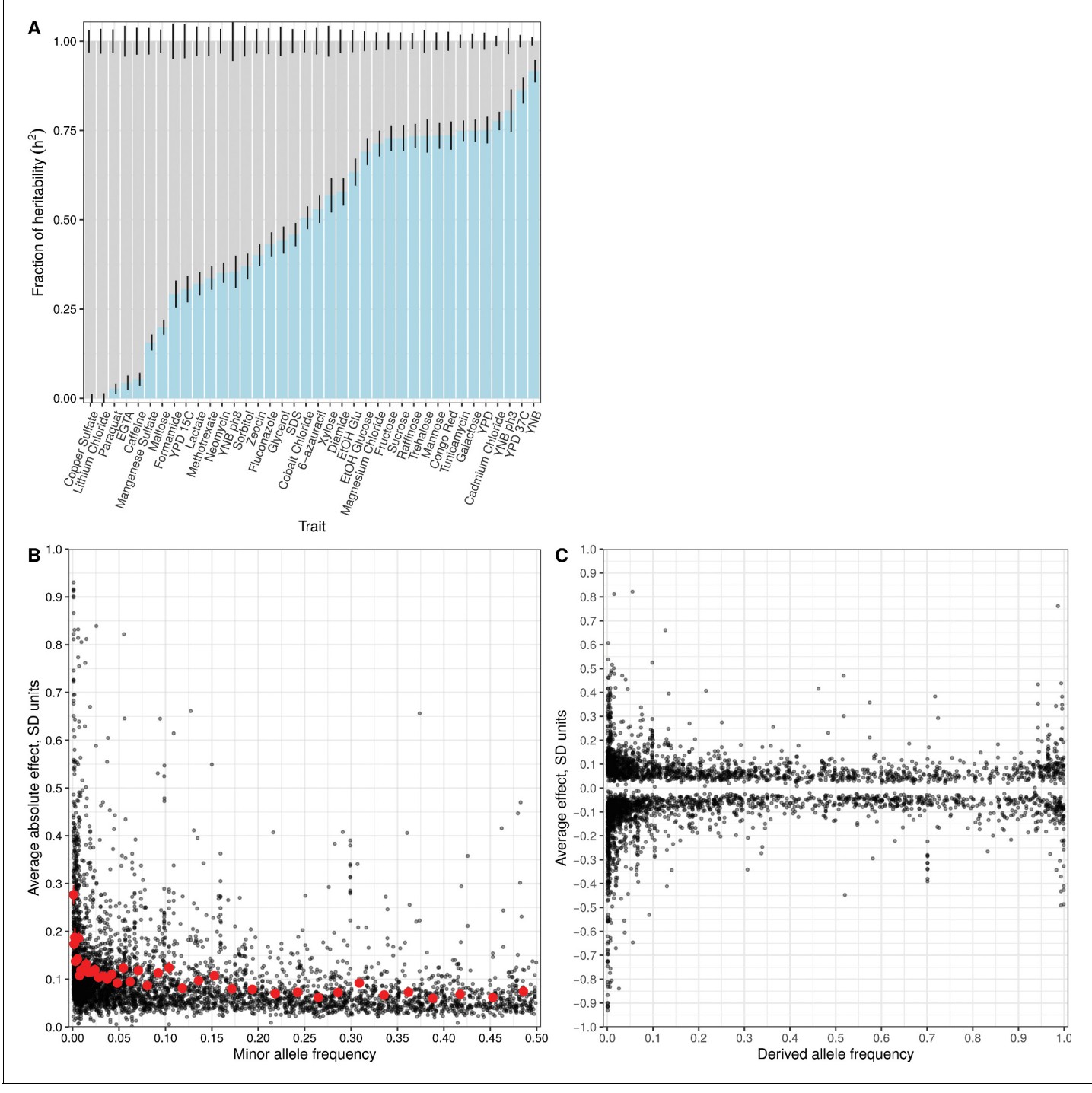

**Figure 3.** Effect size and contribution to trait variation of rare and common variants. (**A**) Stacked bar plots of additive genetic variance explained by rare (blue) and common (gray) variants. Error bars show + /- s.e. (**B**) Minor allele frequency (X-axis) of the lead variant at each QTL (*Peter et al., 2018*) is plotted against QTL effect size (Y-axis). Red points show mean QTL effect sizes for groups of approximately 100 variants binned by allele frequency. Error bars show + /- s.e.m. (**C**) Frequency of the derived allele of each QTL lead variant (X-axis), based on comparison with *S. paradoxus*, is plotted against QTL effect size (Y-axis). Negative values on the Y-axis correspond to variants with effects that are detrimental for growth.

DOI: https://doi.org/10.7554/eLife.49212.006

The following source data and figure supplements are available for figure 3:

**Source data 1.** Detected QTL.
DOI: https://doi.org/10.7554/eLife.49212.012
**Source data 2.** Joint variance component estimates.
*Figure 3 continued on next page*

*Figure 3 continued*

DOI: https://doi.org/10.7554/eLife.49212.013

**Figure supplement 1.** Within-cross variance component analysis.

DOI: https://doi.org/10.7554/eLife.49212.007

**Figure supplement 2.** MAF vs effect size for two simulated architectures.

DOI: https://doi.org/10.7554/eLife.49212.008

**Figure supplement 3.** MAF vs effect size for each trait.

DOI: https://doi.org/10.7554/eLife.49212.009

**Figure supplement 4.** MAF vs unbiased estimate of effect size.

DOI: https://doi.org/10.7554/eLife.49212.010

**Figure supplement 5.** Cumulative genetic variance explained vs minor allele frequency of lead variants.

DOI: https://doi.org/10.7554/eLife.49212.011

An understanding of trait variation at the level of molecular mechanisms requires narrowing QTLs to the underlying causal genes. Such fine-mapping is a challenge because genetic linkage causes variants across an extended region to show mapping signals of similar strength. Statistical fine-mapping aims to address this challenge by estimating the probability that each variant within a QTL region is causal based on the precise pattern of genotype-phenotype correlations (*Farh et al., 2015*; *Pasaniuc and Price, 2017*; *Treusch et al., 2015*). Our crossing design enables us to obtain higher resolution for QTLs observed in two crosses that share a parent strain by looking for consistent inheritance patterns in both. Specifically, we focused on QTLs with effects greater than 0.14 SD units and used a Bayesian framework (*Farh et al., 2015*) to compute the posterior probability that each variant is causal (*Figure 4A*). We then aggregated these probabilities to obtain causality scores for each gene in a QTL. With this approach, we resolved 427 QTLs to single causal genes at an FDR of 20%. Because some QTLs have pleiotropic effects on multiple traits, this gene set contains 195 unique genes, greatly expanding the repertoire of causal genes in yeast. We searched the literature and found that 26 of the 195 genes identified here are supported by previous experimental evidence as causal for yeast trait variation (*Fay, 2013*; *Jerison et al., 2017*; *Sadhu et al., 2016*; *Treusch et al., 2015*; *Wang and Kruglyak, 2014*) (*Figure 4B*; *Figure 4—source data 1*). At a more stringent FDR of 5%, we found 105 unique causal genes, which included 24 of the 26 genes with experimental evidence.

Causal genes were highly enriched for GO terms related to the plasma membrane (45 of 522, 16.5 expected, q = 1.8e-7), metal ion transport (13 of 83, 2.6 expected, q = 0.0009), and positive regulation of nitrogen compound biosynthesis (28 of 393, 12.5 expected, q = 0.0076) (*Figure 4—source data 1*). Strikingly, five of the six genes involved in cAMP biosynthesis were identified as causal (*IRA1*, *IRA2*, *BCY1*, *CYR1*, and *RAS1*; 0.19 expected, q = 0.0002). Additional genes in the RAS/cAMP signaling pathway were also identified as causal, including *GPR1*, which is involved in glucose sensing, *SRV2*, which binds adenylate cyclase, and *RHO3*, which encodes a RAS-like GTPase. In yeast, the RAS/cAMP pathway regulates cell cycle progression, metabolism, and stress resistance (*Tisi et al., 2014*). Variation in many of these genes influenced growth on alternative carbon sources. We hypothesize that the yeast population contains abundant functional variation in genes that regulate the switch from glucose to alternative carbon sources through the RAS/cAMP pathway.

## Discussion

We previously used a cross between lab (BY) and vineyard (RM) strains of yeast to show that the majority of heritable phenotypic differences arise from additive genetic effects, and we were able to detect, at genome-wide significance, specific loci that together account for the majority of quantitative trait variation (*Bloom et al., 2015*; *Bloom et al., 2013*). It has been argued that the BY lab reference strain (commonly known as S288c) used in those and many other yeast studies is genetically and phenotypically atypical compared to other yeast isolates (*Warringer et al., 2011*). Our results here, obtained from crosses among 16 diverse strains, generalize these findings to the *S. cerevisiae*

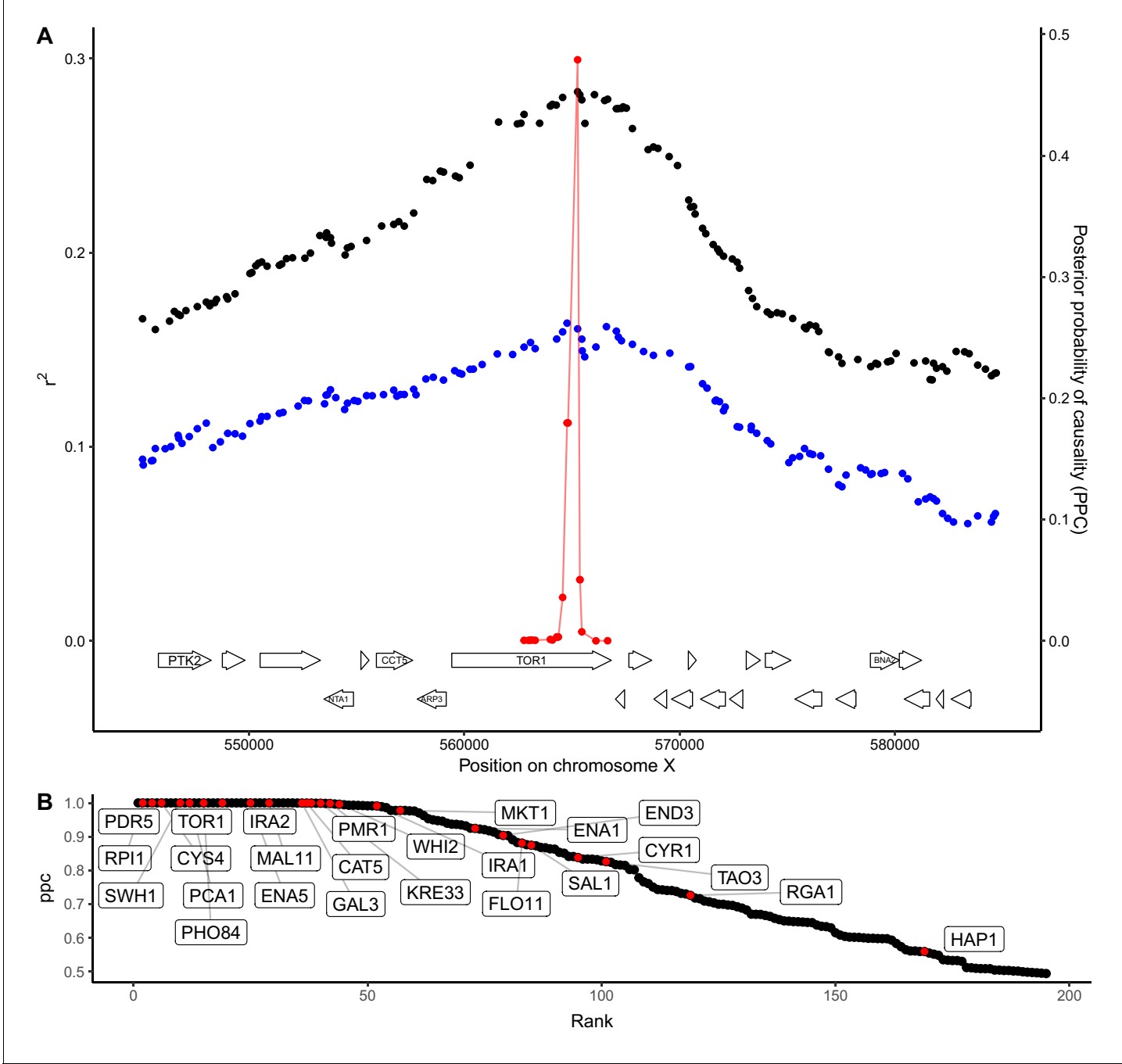

**Figure 4.** QTL fine-mapping at gene-level resolution. (**A**) Statistical fine-mapping of a QTL for growth in the presence of caffeine. Genetic mapping signal, shown as the coefficient of determination between genotype and phenotype (Y-axis, left), is plotted against genome position (X-axis) for crosses between 273614N and YJM981 (black) and YJM981 and CBS2888 (blue). The posterior probability of causality (PPC), plotted in red (Y-axis, right), localizes the QTL to a portion of the gene TOR1. (**B**) PPC is shown as black dots for 195 genes identified as causal at an FDR of 20%, sorted by PPC. Genes containing natural variants that have been experimentally validated as causal for trait variation in prior studies (*Fay, 2013*; *Jerison et al., 2017*; *Sadhu et al., 2016*; *Treusch et al., 2015*; *Wang and Kruglyak, 2014*) are shown in red and labeled with gene names.

DOI: https://doi.org/10.7554/eLife.49212.014

The following source data is available for figure 4:

**Source data 1.** Candidate causal genes and GO enrichments.
DOI: https://doi.org/10.7554/eLife.49212.015

population and show that S288c is not exceptional from the standpoint of genetic variation and quantitative traits. We believe that the findings that the majority of the genetic variance of most traits is additive, and that there is little additive 'missing heritability' in studies with sufficiently large sample sizes, will apply broadly beyond yeast.

We discovered over 4500 quantitative trait loci (QTLs) that influence yeast growth in a wide variety of conditions. These loci likely capture the majority of common variants that segregate in *S. cerevisiae* and have appreciable phenotypic effects on growth, and therefore provide a comprehensive starting point for more fine-grained analyses of the genetic contribution to quantitative trait variation. We were able to localize approximately 8% of the QTLs to single genes based on genetic mapping information alone. Interestingly, these genes cluster in specific functional categories and pathways, suggesting that different strains of *S. cerevisiae* may have evolved different strategies for nutrient sensing and response as a function of specializing in particular environmental niches (*Chantranupong et al., 2015*). In addition to the findings described here, we anticipate that our data set will be a useful resource for further dissecting the genetic basis of trait variation at the gene and variant level (*Peltier et al., 2019*), and for evaluating statistical methods aimed at inferring causal genes and variants. In particular, the set of loci and genes identified here provides an ideal starting point for massively parallel editing experiments that directly test the phenotypic consequences of sequence variants (*Shendure and Fields, 2016*).

By combining our results with deep population sequencing in yeast (*Peter et al., 2018*), we were able to examine the contributions of variants in different frequency classes to trait variation. This approach avoids statistical confounding between variant frequency and effect size that occurs when both are measured in the same study sample. We observed a broad range of genetic architectures across the traits studied here, with variation in some traits dominated by common variants, while variation in others is mostly explained by rare variants. Overall, rare variants made a disproportionate contribution to trait variation as a consequence of their larger effect sizes. A complementary mapping approach in an overlapping set of yeast isolates also revealed enrichment of rare variants with larger effects (*Fournier et al., 2019*). These results are consistent with the finding from GWAS that common variants have small effects, as well as with linkage studies that find rare variants with large effect sizes. Our study design also revealed a substantial component of genetic variation—variants with low allele frequency and small effect size—that has been refractory to discovery in humans because both GWAS and linkage studies lack statistical power to detect this class of variants. Recent work in humans has suggested that rare variants account for a substantial fraction of heritability of complex traits and diseases (*Wainschtein et al., 2019*). Our study presents a more direct and fine-grained view of this component of trait variation and implies that larger sample sizes and more complete genotype information will be needed for more comprehensive studies in other systems.

## Materials and methods

### Data availability

Unless otherwise specified, all computational analyses were performed in R (v3.4.4). Analysis code and processing scripts are available at https://github.com/joshsbloom/yeast-16-parents (*Bloom, 2019*; copy archived at https://github.com/elifesciences-publications/yeast-16-parents). Additional links to generated data are also provided in the github repository. The version numbers of R packages used are listed in this repository. Sequencing data for parents and segregants is available in the Sequenced Read Archive (SRA) under the Bioproject ID PRJNA549760.

### Short-read and synthetic long read sequencing of parental strains

Parental genotypes were obtained by deep (>100X) paired-end sequencing of the 16 parental strains. A VCF file containing SNPs and small indels was generated for the parents using bwa (v0.7.1) (*Li, 2013*) to align to the sacCer3 reference (*Engel et al., 2014*), Picard (v2.12.2) (*Broad Institute, 2019*) to remove PCR duplicates, and the GATK HaplotypeCaller (v3.8) (*Van der Auwera et al., 2013*) with expected sample ploidy set to 1. A separate pipeline was developed to leverage additional synthetic long-reads (Illumina/Moleculo) to identify larger structural variants in the parents. Briefly, synthetic long-read assemblies were filtered to only include scaffolds greater than 10 kb. Scaffolds were corrected with our short-read data using Pilon (*Walker et al., 2014*).

CNVs were discovered using custom scripts modified from scripts originally used to generate calls for testing LUMPY (*Layer et al., 2014*). CNVs were genotyped in all parents using the approach presented in SVTyper (*Ebler et al., 2017*). Scripts associated with the CNV detection pipeline are available at https://github.com/theboocock/long_read_cnv (*Boocock, 2019*; https://github.com/elifesciences-publications/long_read_cnv).

## Construction of haploid segregant panels

Segregants for the BY-RM cross and YPS163-YJM145 cross were obtained by sporulation of the hybrid diploid parents for 5–7 days in SPO++ sporulation medium (http://dunham.gs.washington.edu/sporulationdissection.htm) and tetrad dissection using the MSM 400 dissection microscope (Singer Instrument Company Ltd.). Four-spore tetrads were retained. For BYxRM, one segregant was randomly chosen per tetrad (*Bloom et al., 2013*). For the YPS163-YJM145 cross, all segregants from ~250 tetrads were used. For all other crosses, the hybrid diploids were were pre-grown in YPD with either G418 or cloNat, depending on which fluorescent magic marker plasmid they contained (*Treusch et al., 2015*). Then they were sporulated in SPO++ and either cloNat or G418 for 5–7 days. A random spore prep was used to isolate haploid progeny (https://openwetware.org/wiki/McClean:Random_Spore_Prep), modified to exclude the use of glass beads for spore separation. Cells were plated on selective media, grown for two days, and colony fluorescence was visualized. Green fluorescent colonies and red fluorescent colonies corresponding to MATa and MATα haploid progeny were picked to deep-well 96-well plates and then split into frozen stocks.

## Preparation of whole-genome sequencing libraries for segregants

Yeast were grown in 1 ml of yeast peptone dextrose in 2 ml deep-well 96-well plates (Thermo Scientific). Plates were sealed with Breathe-Easy gas-permeable membranes (Sigma-Aldrich). Yeasts were grown without shaking for 2 days in a 30°C incubator. Cell walls were digested with Zymolase, and DNA was extracted using either the 96-well DNeasy Blood and Tissue kits (Qiagen) for the BY-RM and YPS163-YJM145 segregants, or 96-well E-Z 96 Tissue DNA kit, following the bacterial protocol (Omega) for all other segregants. DNA concentrations were determined using the Quant-iT dsDNA High-Sensitivity DNA quantification kit (Invitrogen) and the Bio-Tek Synergy two-plate reader. DNA was diluted to 0.22 ng per µl using a Biomek FX liquid handing robot (Beckamn Coulter). For each segregant, 5 µl of 0.22 ng per µl of DNA was added to 4 µl of 5X Nextera HWM buffer (Illumina), 6 µl of water and 5 µl of 1/35 diluted Nextera enzyme. The transposition reaction was performed for 5 min at 55°C. Directly after the tagmentation reaction and without additional sample purification, Illumina sequencing adaptors and custom indices were added by PCR. 10 µl of tagmented DNA was combined with 0.5 µl each of 10 µM index primers (one of N701-N712 plus one of 96 custom indices, see *Supplementary file 1*), 5 µl of 10X Ex Taq buffer, 0.375 µl Ex Taq polymerase (Takara), 4 µl of 2.5 mM dNTPs and 29.625 µl of water, and amplified with 20 cycles of PCR. Up to 1152-plex libraries were run on a Hiseq 2500 with single end 150 bp reads, except BY-RM (*Bloom et al., 2013*) and YPS163-YJM145 which were sequenced with 100 bp reads.

## Segregant genotype calling

Fastq files for were demultiplexed using fastq-multx (v1.3.1) (*Aronesty, 2013*) and aligned to the SacCer3 version of the reference genome using bwa. Adapter sequences were trimmed from reads and Phred33 quality scores were computed with Trimmomatic (v0.32) (*Bolger et al., 2014*). PCR duplicates were removed using Picard and then merged into one CRAM file per cross using Picard. VCF files were generated for each cross using the GATK haplotype caller (*Van der Auwera et al., 2013*) and genotypes were called at known variant sites between the parental strains. Additional custom provided R code was used to remove regions with strong mapping bias toward the reference genome (*Albert et al., 2018*), filter poor quality markers, and remove segregants with too many crossovers, likely corresponding to diploid contaminants. Missing segregant genotype information was imputed using a hidden Markov model (HMM) implemented in R/QTL (*Arends et al., 2010*). Structural variants identified in the parent VCF files were considered missing information in the segregants and the HMM was used to impute genotypes at those sites.

## Phenotyping by endpoint colony growth

Segregants were arrayed to 384-well liquid plates in duplicate with different plate positions across the duplicates. Segregants were grown in YPD for approximately 48 hr without shaking and then pinned to agar plates using a BM-5 colony arraying robot (S and P Robotics). Plates were incubated for 48 hr and end-point growth was quantified by automated plate imaging using the colony arraying robot. Colony radii were calculated using functions in the EBImage R package (*Pau et al., 2010*), and endpoint growth measurements were filtered and normalized for plate effects as described previously (*Bloom et al., 2015*; *Bloom et al., 2013*). In addition, a manual filtering step was used to filter out aberrant colonies arising from technical artefacts, such as from wet spots on the agar plates at the time colonies were pinned. Unless otherwise specified, the average value across replicates was used per segregant for all downstream analyses.

## Within-cross QTL mapping

QTL were mapped using a forward stepwise regression procedure that controls the FDR (*G'Sell et al., 2013*) for each trait and cross. We tested for linkage at each marker along the genome by calculating $r^2$, where r is the Pearson correlation coefficient between segregant genotypes at the marker and segregant phenotypes. 10,000 permutations of phenotype to strain assignment were performed and this statistic was calculated across the genome for each of the permutations. For each of the permutations, the maximum statistic was recorded to generate an empirical null distribution of the maximum statistic (*Churchill and Doerge, 1994*). A p-value was calculated as the probability the observed maximum statistic comes from the empirical null distribution of maximum statistics. If the observed statistic was greater than all of the empirical null statistics the p-value was recorded as 1e-4. The p-value was added to a set of p-values ($p_1, \ldots p_k$), and the entire procedure was repeated (including permutations) with the previously identified marker(s) included as regression covariates. A 'FowardStop', FDR-controlling statistic (*G'Sell et al., 2013*) was calculated as $-\frac{1}{k}\sum_{i=1}^{k}\log(1 - p_i)$. We continued to add selected markers to a multiple regression model as long as the 'ForwardStop' statistic was less than or equal to 5%.

We note that we chose to use this procedure rather than procedures we have used in the past (*Albert et al., 2018*; *Bloom et al., 2015*; *Bloom et al., 2013*) because it is simple, does not require exchangeability of statistics across different traits, gives very similar results as previous methods, and we verified through simulations (not shown) that it controls FDR for forward stepwise model selection under different QTL architectures.

For this within-cross QTL mapping procedure, we re-localized QTL peak positions for QTL detected by the forward selection procedure. Specifically, for each QTL peak we included all other detected QTL peaks (as detected from the forward selection procedure) as covariates in a multiple regression model, and scanned each marker on the chromosome on which the QTL peak being re-localized was detected to identify the marker that maximized the likelihood of the multiple regression model. The marker that maximized the likelihood of the multiple regression model was retained as the new, re-localized, QTL peak position.

## Cross-validation procedures to estimate heritability explained by QTL

The amount of additive variance explained by detected QTLs was estimated using cross-validation. For the within-cross analysis, segregants were randomly split into 10 sets. Each set of segregants was left out of the procedure one at a time (held-out set). The within-cross QTL mapping procedure was performed for all the other sets (training set). For the QTL markers detected in this training set and with effects estimated in the training set, the amount of variance explained by the joint model of the set of significant QTL markers was estimated in the held out set. For the joint analysis described below, we performed a similar procedure, splitting the segregants within each cross into 10 sets, leaving one of the sets from each cross out (held-out set) identifying QTL jointly across the other sets (training set) and estimating their effects in each cross (training set) and then estimating the variance explained in the held-out set.

## Within-cross analysis to estimate additive and pairwise genetic interaction variance

To estimate the fraction of phenotypic variance attributable to additive genetic effects for each cross and trait we fit the model $y = a+e$, where $y$ contains the segregant phenotype values and is standardized to have mean 0 and variance 1. Here, $a$ are the additive genetic effects and the residual error is denoted as $e$. The distributions of these effects are assumed to be multivariate normal with mean zero and variance-covariance as follows:

$$a \sim N(0, \sigma_A^2 A) \ \ and \ e \sim N(0, \sigma_{EV}^2 I)$$

Here, $A$ is the additive relatedness matrix, the fraction of genome shared between pairs of segregants and was calculated as $MM'/n$ where $M$ is the $n$ x $m$ matrix of standardized marker genotypes, $n$ is the number of segregants and $m$ is the number of markers.

We also fit an expanded model to estimate the relative contribution of additive vs non-additive (pairwise epistatic) effects. For the pairwise epistatic component, we believe that the assumption that all pairs of loci contribute to trait variation with effect sizes drawn from a single normal distribution is violated when one or a few QTL-QTL interactions with large effects are present, resulting in a downward bias. We previously showed (*Bloom et al., 2015*) that loci involved in such stronger interactions can be detected in additive scans. Therefore, by explicitly including additive QTLs in the three components model, we avoid making the assumption that the effect sizes of all locus pairs are drawn from the same normal distribution and obtain a better estimator of total two-way epistatic variance when large-effect QTL-QTL interactions are present. This model was parameterized as:

$$y = \beta X + Zq + Za + Zf + Zg + Zi + Zp + e$$

The distributions of these effects are assumed to be multivariate normal with mean zero and variance-covariance as follows:

$$q \sim N\left(0, \sigma_{A_{QTL}}^2 A_{QTL}\right), \ a \sim N(0, \sigma_A^2 A), \ f \sim N\left(0, \ \sigma_{A_{QTL}*A_{QTL}}^2 A_{QTL} \circ A_{QTL}\right),$$

$g \sim N\left(0, \sigma_{A_{QTL}*A}^2 A_{QTL} \circ A\right), i \sim N(0, \sigma_{A*A}^2 A \circ A), p \sim N(0, \sigma_R^2 I_n),$ and $e \sim N(0, \sigma_{EV}^2 I_m)$ where $y$ is a vector of length $L$ that contains phenotypes for n segregants including replicate measurements such that $L$ = n x [number of replicates]. $\beta$ is a vector of estimated fixed effect coefficients. $X$ is a matrix of fixed effects (here $\beta$ is the overall mean, and $X$ is a $1_L$ vector of ones unless otherwise specified). $Z$ is an $L$ x n incidence matrix that maps $L$ total measures to $n$ total segregants. In order, the random effect terms correspond to the effects of detected QTL, effects from the whole genome, epistatic interactions between detected QTL, epistatic interactions between additive QTL and the genome, epistatic interactions between all pairs of markers across the genome, and residual repeatability, following very similar methods and syntax as described previously (*Bloom et al., 2015*). We also fit a model that omitted the terms for epistatic interactions between detected QTL, and epistatic interactions between additive QTL and the genome. The mixed model was fit with the regress R package (*Clifford and McCullagh, 2014*) using restricted maximum likelihood estimation (REML). Standard errors of variance component estimates were calculated as the square root of the diagonal of the Fisher information matrix from the iteration at convergence of the Newton-Raphson algorithm. These procedures were used for all other mixed model analyses described below. For the analysis that compared the fraction non-additive to additive variation, we calculated $\frac{\sigma_{A_{QTL}*A_{QTL}}^2 + \sigma_{A_{QTL}*A}^2 + \sigma_{A*A}^2}{\sigma_{A_{QTL}}^2 + \sigma_A^2}$.

## Allele-frequency lookup in 1011 yeast isolate population

We used bcftools isec (*Li et al., 2009*) to intersect our VCF containing sequence variant information on the 16 parental strains with the 1011 yeast isolate VCF generated by *Peter et al. (2018)*, and vcftools (*Danecek et al., 2011*) to further filter only biallelic variants. This subset of 259,647 biallelic markers was used for variance components analysis and joint QTL mapping across the panel. Allele frequencies in the larger panel of 1011 yeast isolates were extracted from the provided VCF (*Peter et al., 2018*). Derived allele frequencies were calculated by using nucmer (*Marçais et al., 2018*) to perform whole genome alignment between the sacCer3 reference assembly and the

CBS432 assembly of *S. paradoxus*. Variants were identified using delta-filter and show-snps commands provided in nucmer. Biallelic variants in our panel were classified as ancient if the variant matches the *S. paradoxus* sequence and recent if not. The unfolded allele frequency was calculated as the frequency of the recent variant. We could determine ancestral status for approximately 80% of the biallelic variants. To improve power for enrichment tests, we used derived allele frequency <5% and >95% as cutoffs when comparing effect sizes and signs of effects between derived and ancestral variants.

### Genotype recoding for joint analyses

We coded the biallelic markers for which we had allele frequency data from the larger yeast isolate panel as −1 for matching the reference strain, or one if not matching the reference. If a variant does not segregate in a particular cross it was treated as missing in that cross.

### Mixed model analysis with allele-frequency partitioning

We fit the following mixed model model per trait (jointly across the different crosses):

$$y = \beta X + r + c + e$$

The distributions of these effects are assumed to be multivariate normal with mean zero and variance-covariance as follows:

$r \sim N(0, \sigma_R^2 A_{maf<1\%})$, $c \sim N(0, \sigma_C^2 A_{maf>=1\%})$, *and* $e \sim N(0, \sigma_{EV}^2 I_m)$ where $y$ is a vector of length 13,950 that contains phenotypes for segregants concatenated across the different crosses. β is a vector of estimated fixed effects of each cross. **X** is an incidence matrix mapping segregants to crosses. Here, the two relatedness matrices $A_{maf<1\%}$ and $A_{maf>=1\%}$ were calculated separately for all markers with MAF<1% and MAF>=1% respectively in the larger panel of 1,011 yeast isolates. Per marker, the genotype values were scaled to have mean 0 and variance 1, for each of the segregants from crosses in which that marker segregates. Markers that are fixed within a cross were excluded from the subsequent calculation of genetic covariance. The rationale for excluding data for variants not segregating in a given cross is that all such variants are completely confounded with each other and with any other effects specific to that cross. Thus, their effects are more appropriately captured by including a fixed effect for each cross within the analysis. Then, with **M** being the n segregants by m markers matrix corresponding to the standardized genotypes for that subset of markers, we calculated the relatedness matrix as a Gower's centered matrix (*Forni et al., 2011*; *Kang et al., 2010*; *McArdle and Anderson, 2001*) $\frac{MM'}{\frac{tr(MM')}{n}}$ which has the property that the average diagonal coefficient equals 1.

We used the same logic to construct additional covariance matrices when more finely binning variants by allele frequency in the external panel (seven allele-frequency bins model). Bins were chosen to contain approximately equal numbers of variants. We also fit the seven allele-frequency bins model using only variants that were private to each parent (variants that only segregate in a pair of crosses). In this last model, the allele-frequency of variants used for the analysis are all approximately the same across the panel. Therefore, this last model does not make the assumption that the variance of variants effects is inversely proportional to their frequencies in the mapping panel (*Yang et al., 2010*).

The procedure for fitting these models was the same as described above in the section 'within-cross variance component analysis'.

### Accounting for large effect QTL and polygenic background for all chromosomes except the chromosome of interest for joint QTL mapping

For each chromosome of interest and for each trait and cross and trait, we calculated $y_c = CQ + a_L + s_c$ where $y_c$ is the vector of trait values for a given trait and cross, **Q** is a matrix of QTL genotypes at peak markers from the within-cross mapping described above, with FDR < 5% that are not located on the chromosome of interest, $C$ is a vector of estimated QTL effects from the section 'within-cross QTL mapping', $a_L$ is the additive genetic variance from all chromosomes excluding the chromosome of interest. $a_L$ comes from the REML-based BLUP estimate of the effect all other chromosomes,

including the fixed effects of detected QTL on the other. The goal of this step was to obtain the residual trait values $s_c$ that can be used to scan for QTLs on a chromosome of interest and corrects for mapped genetic sources of variation that do not arise from the chromosome of interest (*Yang et al., 2014*).

## Joint QTL mapping

Under the assumption that a causal biallelic variant has a consistent additive effect in all the crosses in which it segregates, we implemented a model to identify such variants jointly across our entire segregant panel (*McMullen et al., 2009*; *Stich, 2009*). This procedure increases statistical power. For example, for variants that are private to one of the 16 parental strains, this procedure will approximately double the observed number of instances of the minor allele, resulting in less noisy estimates of variant effects. For variants that are shared among multiple parents, the increase in the observed number of instances of the minor allele will be greater.

For each trait and each chromosome, and then for each marker on that chromosome, we calculated a t-statistic as $\frac{r}{\sqrt{\frac{1-r^2}{n-2}}}$. Here, r is the Pearson correlation between the recoded segregant genotypes across the panel, and the vector s, which corresponds to the values of $s_c$ described in the previous section concatenated across the different crosses. The number of informative segregants, n, differs for each biallelic variant, and corresponds to the sum of the sample sizes for each cross in which the variant segregates. P-values were calculated that factor in the different number of informative segregants, n, in the calculation of the degrees of freedom using built-in R functions. The -log10 (p) was recorded. This statistic was calculated for each marker on the chromosome. 1000 permutations of phenotype to strain assignment were performed, but these permutations were performed with phenotype values within each cross (we did not permute values between crosses) and this statistic was calculated across the genome for each of the permutations. For each of the permutations, the maximum statistic was recorded to generate an empirical null distribution of the maximum statistic (*Churchill and Doerge, 1994*). A new corrected p-value was calculated as the probability the observed maximum statistic comes from the empirical null distribution of maximum statistics. If the observed maximum statistic was greater than all of the empirical null maximum statistics the p-value was recorded as 1e-3. The p-value was added to a set of p-values ($p_1, \ldots p_k$), and the entire procedure was repeated (including permutations) with the previously identified marker(s) included as regression covariates. A 'FowardStop', FDR-controlling statistic (*G'Sell et al., 2013*) was calculated as described above. We continued to add selected markers to a multiple regression model as long as the 'ForwardStop' statistic was less than or equal to 5%.

## Effect size estimation for joint QTL mapping

The peak markers (lead variants) from this procedure were then used for effect size estimation. For each trait and cross, the phenotypes are scaled to have mean 0 and variance 1, and effect sizes within each cross are estimated using multiple regression for the peak markers that segregate within that cross. The betas in this analysis correspond to the differences in the means between the two QTL alleles (conditional on the effects of the other segregating QTL). For peak markers that segregate in multiple crosses, the average betas over the different crosses are shown in *Figure 3*. Unbiased estimates of QTL effect size (*Figure 3—figure supplement 4*) were obtained by the same procedure except peak detection was performed in 9/10 of the data and effects estimated in the 1/10 of the data left out. Allele frequencies of the lead variants were looked up in the 1011 isolate panel.

## Statistical fine-mapping to identify causal genes

We implemented the probabilistic identification of causal snps (PICS) procedure, a Bayesian approach to estimate the probability that a variant is causal. A very thorough description of the method, including details about the logic and implementation, is present in *Farh et al. (2015)*. We aggregated these probabilities within genes to estimate the probability that a gene contains the causal variant. We noted the position of the observed QTL peak (called the 'lead' variant in the GWAS literature), and its effect size for all QTL that explained more than 2% of phenotypic variance from the within-cross mapping (equivalent to 0.1414 SD units). We assumed that the prior probabilities of a variant being causal, or being identified as a lead variant, are equal. For this analysis, we

only used variants that fall within a 50 kb window centered around the detected QTL peak. For each variant within this window, we simulated the observed QTL effect size on the background of noise, 500 times. Here, noise was estimated as the residual error of the within-cross QTL model for that trait and cross. Each of the simulations was generated by a different permutation of the assignment of the residual error to segregant. We then repeated our mapping procedure for the simulated data and calculated the fraction of simulations where the observed QTL peak from our trait mappings was the lead variant given the simulated causal variant. This posterior probability was estimated for each of the variants within the 50 kb window, and then normalized so that the sum of all the probabilities in the window is 1. This generated a variant-level probability of causality for each variant within the window for that trait and cross.

Next, we identified overlapping QTL. Overlapping QTL were defined as the QTL coming from neighboring crosses that shared a parent, have 1.5 LOD drop confidence intervals that overlap, and have QTL effect directions that are consistent between the neighboring crosses. For these overlapping QTL, we calculated the product of the causality probabilities (described above) for each variant shared between the two crosses (and segregating in both crosses) and then normalized these probabilities so that they sum to 1. To calculate the probability that a gene was causal, we summed these probabilities for all variants that fell within each gene. Here, a gene was defined as all variants that fell within the defined open-reading frame as well as variants that fell halfway between the start and stop of the adjacent open-reading frames. We calculated a FDR by sorting the observed posterior probabilities of causality per gene from highest to lowest, calculating a posterior error probability as one minus the posterior probability of causality, and calculating the cumulative mean of these probabilities (*Käll et al., 2008*; *Storey, 2003*; *Storey and Tibshirani, 2003*).

We note that the causal gene statistic is an estimate of the posterior probability that a gene is causal assuming that one causal variant in the defined window is responsible for generating a signal in two crosses that share a parent strain, that we estimate the effects of causal variants in both crosses without error, and that genotypes are called without error.

## Gene ontology enrichment analyses

We tested for GO enrichments using the R package topGO (*Alexa and Rahnenfuhrer, 2018*), using the Fisher test for enrichment ant the 'classic' scoring method that does not adjust the enrichments for significance of child GO terms.

## Acknowledgements

We thank Bogdan Pasaniuc, Frank W Albert, Olga T Schubert, Liangke Gou, Tzitziki Lemus Vergara, Matthieu Delcourt, Longhua Guo, and Eyal Ben-David for helpful manuscript feedback and edits. We thank Illumina for performing synthetic long-read sequencing of the parental yeast strains. This work was supported by funding from the Howard Hughes Medical Institute (to LK) and NIH grant R01GM102308 (to LK). The authors declare no competing financial interests.

## Additional information

### Competing interests

Sebastian Treusch: Sebastian Treusch is now affiliated with Intrexon, although all work for this study was carried out while ST was affiliated with UCLA. The author has no other competing interests to declare. The other authors declare that no competing interests exist.

### Funding

| Funder | Grant reference number | Author |
|---|---|---|
| National Institutes of Health | R01GM102308 | Joshua S Bloom<br>Meru J Sadhu<br>Laura Day<br>Holly Oates-Barker<br>Leonid Kruglyak |

Howard Hughes Medical Insti-
tute

Joshua S Bloom
Laura Day
Holly Oates-Barker
Leonid Kruglyak

The funders had no role in study design, data collection and interpretation, or the decision to submit the work for publication.

## Author contributions
Joshua S Bloom, Conceptualization, Resources, Data curation, Software, Formal analysis, Supervision, Validation, Investigation, Visualization, Methodology, Writing—original draft, Writing—review and editing; James Boocock, Data curation, Software, Formal analysis, Writing—review and editing; Sebastian Treusch, Conceptualization, Resources; Meru J Sadhu, Investigation, Writing—review and editing; Laura Day, Holly Oates-Barker, Resources, Methodology; Leonid Kruglyak, Conceptualization, Supervision, Funding acquisition, Writing—original draft, Project administration, Writing—review and editing

## Author ORCIDs
Joshua S Bloom (iD) https://orcid.org/0000-0002-7241-1648
Leonid Kruglyak (iD) https://orcid.org/0000-0002-8065-3057

## Decision letter and Author response
Decision letter https://doi.org/10.7554/eLife.49212.023
Author response https://doi.org/10.7554/eLife.49212.024

# Additional files

## Supplementary files
• Supplementary file 1. Key resources table.
DOI: https://doi.org/10.7554/eLife.49212.016

• Supplementary file 2. Barplots depicting results from within-cross variance component analyses. For each trait and cross, a stacked barplot is shown representing the results from a multiple variance component model. Phenotypic covariance was modeled as the sum of QTL effects (light blue, denoted as Q), additive genome effects (dark blue, denoted as A), interactions between additive QTLs (dark green, denoted as Q∘Q), interactions between additive QTLs and the rest of the genome (light green, denoted as Q∘A), interactions between all loci in the genome (sea green, denoted as A∘A), residual effect of strain (pink), and residual error (white).
DOI: https://doi.org/10.7554/eLife.49212.017

• Supplementary file 3. QTL mapping results for each trait and cross. Results from QTL mapping are shown for each trait. Each subpanel represents the results for a cross. Along the Y-axis the two parental strains for each cross are shown. Position of QTL along the genome is represented on the X-axis. The arrows represent QTL effects from the within-cross mapping. The arrows point toward the strain that increases growth. The size of the arrow reflects the magnitude of the QTL effect. Full length arrows represent QTL that explain more than 25% of phenotypic variance,  length arrows represent QTL that explain between 8% and 25% of phenotypic variance, ½ length arrows represent QTL that explain between 4% and 8% of phenotypic variance, and short arrows represent QTL that explain less than 4% of phenotypic variance. Large effect QTL (explaining more than 4% of phenotypic variance) are colored black, and small effect QTL (less than 4% of phenotypic variance) are colored gray. The green vertical lines correspond to QTL detected from the joint QTL mapping analysis (Materials and methods).
DOI: https://doi.org/10.7554/eLife.49212.018

• Transparent reporting form DOI: https://doi.org/10.7554/eLife.49212.019

## Data availability
Unless otherwise specified, all computational analyses were performed in R (v3.4.4). Analysis code and processing scripts are available at https://github.com/joshsbloom/yeast-16-parents (copy

archived at https://github.com/elifesciences-publications/yeast-16-parents). Additional links to generated data are also provided in the github repository. The version numbers of R packages used are listed in this repository. Sequencing data has been deposited in the SRA under the accession code PRJNA549760.

The following dataset was generated:

| Author(s) | Year | Dataset title | Dataset URL | Database and Identifier |
|---|---|---|---|---|
| Bloom JS, Boocock J, Treusch S, Sadhu MJ, Day L, Oates-Barker H, Kruglyak L | 2019 | Rare variants contribute disproportionately to quantitative trait variation in yeast, Jun 19 '19 | http://www.ncbi.nlm.nih.gov/bioproject/549760 | SRA Bioproject, PRJNA549760 |

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
