## [Decision Letter]

**Acceptance summary:**

The authors show that rare and recent variants contribute proportionally more than common ones to trait variation in budding yeast. They combine a powerful quantitative genetics approach with extensive trait phenotyping and population genomics data to illustrate how natural selection may be keeping mutations with large effects from reaching high frequency. The expected negative correlation between effect size and frequency has been studied theoretically, but much less so at the experimental level because of the complex type of data needed. The study has broad significance, from population genetics, to quantitative genetics and evolution of complex traits.

**Decision letter after peer review:**

Thank you for submitting your article "Rare variants contribute disproportionately to quantitative trait variation in yeast" for consideration by *eLife*. Your article has been reviewed by three peer reviewers, and the evaluation has been overseen by a Reviewing Editor and Naama Barkai as the Senior Editor. The reviewers have opted to remain anonymous.

The reviewers have discussed the reviews with one another and the Reviewing Editor has drafted this decision to help you prepare a revised submission.

Summary:

Your manuscript addresses an important question in genetics and evolutionary biology: what are the relative contributions of genetic variants to phenotypic variation and do these contributions correlate with the frequency of these variants within a species? The reviewers agree that this is an important question that has rarely been comprehensively addressed. They therefore find the work of interest and the findings to be important for the field, in addition to appreciating the quality of the writing and of the presentation. However, they identify some aspects that would need to be reconsidered or better presented and interpreted (details below). In addition, your paper could be strengthened if it was extended to include a more detailed Introduction on why these questions are important for the non-specialists. You could also include more discussions on the implications of your findings, including the implications for biology and evolution in general and for yeast in particular. Since *eLife* is a generalist journal, the manuscript would appeal to a larger audience with these changes. Your manuscript is currently very short so this would be feasible by the addition of a few short sections in the current structure of the manuscript. You will see that one specific comment relates to the novelty of the work with respect to previous studies. This means that the novelty may not be obvious as presented. It would therefore also be important to emphasize this aspect in a revised version.

Essential revisions:

1) There are two analyses, within cross and joint analysis. I have to go back and forth between Results and Materials and methods to figure out exactly what is done. It would be nice to make clear when discussing the results from which one they are derived.

2) Because the segregants are haploid, there is only the A x A interaction. The majority of variance generated by A x A in fact goes into the additive variance, hence the non-additive variance is small. The authors did not make a big deal out of the fact that non-additive variance is only 1/6 of additive variance, but I feel it's important to stress that large additive variance is expected given the population design. In addition, when estimating the variance attributable additive variance and epistatic variance, the authors broke the non-additive variance into three components, AQTL x A, AQTL x AQTL, and A x A. I wonder if this is necessary because there was no mention of the differences between these three components. A single component A(all) x A(all) could be fitted.

3) The two-component mixed model analysis has some caveats. There is correlation between rare and common variants, i.e., the variance components are not orthogonal. This makes any claim about the relative importance of rare versus common less reliable. For example, for the Cadmium chloride trait (Figure 3—figure supplement 1), the 7-component model seems to disagree with the 2-component model, with MAF < 0.01 explaining much less in the 7-component model than in the 2-component model in A. I think comparing the 2-component and one component model could suffer from the same problem. Perhaps a more appropriate (but still not perfect) analysis is to fit single component model first and then fit two component models, do it in two sequential orders (rare, then rare + common; versus common, then common + rare), and look at how the cumulative variance increase. This will tell you which MAF class is more important or can explain more variance. I think in this case, one-component model is more informative than the 2-component model.

4) In Figure 3, the relationship between effects and MAF or DAF is a major result. Although many other papers have reported similar results, I think this paper (and the co-submitted paper from Fournier et al.) has the most appropriate design, i.e., the discovery panel is independent of where MAF is estimated. Given its central role in this paper, it probably deserves a bit more clarity. A few questions came to mind when reading this part of the paper. In what analysis is the effect size estimated, single crosses or joint? Could you briefly explain how the effect sizes are estimated in the Results section? If effects estimated from joint analysis, the t-statistic used a factor (n-2) to normalize the degree of freedom, which is smaller for rare variants. This would lead to Beavis effect. I believe the authors used a cross validation strategy to estimate effects, but it's not very clear by reading the Materials and methods. Can you also plot 2pqa^2 versus MAF? Even if a is large, the variance contributed by rare variants could be small.

5) The main conclusion of the manuscript is that rare variants significantly contribute to genetic variance. In my view, this conclusion is biased as these rare causal variants are being analysed in genetic backgrounds in which they are no longer rare; actually, these variants are biallelic. Several studies have shown that a rare variant of MKT1(89A) is a significant contributor to phenotypic variation whenever it is present in segregating populations. However, MKT1(89A) allele hardly identified when one of the parents is not S288c, the strain which harbours this allele. So, the extension that if the rare variant has a significant effect in a sub-population, then its effect size would be similar in a large heterogeneous population is false. Furthermore, the authors conclude that in their larger 16 strain segregant populations, a representative distribution of 1000 strain collection, most of the variants have additive effects. This the authors claim is revalidation of their other previous studies (Bloom et al., 2013, 2015), where they identified that most of the causal variants between BYxRM had additive effects. However, in their subsequent paper (Frosberg et al., 2017, PMID 28250458) and another paper (Yadav et al., 2016, PMID 28172852) showed that variance mapping in BYxRM segregants helped to account for genetic interactions and showed how non-additive interactions also contribute significantly to phenotypic variation. Therefore, I find that just doing a few more strains and larger no. of segregants per cross does not make this manuscript a significant advance over the previous studies. One can argue that taking into account all causal variants identified till date (Fay, 2013), one can identify what frequency of rare variants have been identified, e.g. a typical example being MKT1(89A) allele as causal, even though their effect size will not be identified using this strategy. Peltier et al., 2019, show that 284 rare QTNs variants have been identified till date and these functional variants being private to a subpopulation, possibly due to their adaptive role to a specific environment. Moreover, this conclusion can be made without these extensive experimental crosses.

---

## [Author Response]

Essential revisions:1) There are two analyses, within cross and joint analysis. I have to go back and forth between Results and Materials and methods to figure out exactly what is done. It would be nice to make clear when discussing the results from which one they are derived.

We apologize for any confusion resulting from our presentation of these analyses. We have made a clarification to the text to emphasize that, except for the one paragraph comparing the two analyses beginning with “We complemented the joint analysis with QTL mapping within each cross…”), the text focuses entirely on results from the joint analysis.

2) Because the segregants are haploid, there is only the A x A interaction. The majority of variance generated by A x A in fact goes into the additive variance, hence the non-additive variance is small. The authors did not make a big deal out of the fact that non-additive variance is only 1/6 of additive variance, but I feel it’s important to stress that large additive variance is expected given the population design. In addition, when estimating the variance attributable additive variance and epistatic variance, the authors broke the non-additive variance into three components, AQTL x A, AQTL x AQTL, and A x A. I wonder if this is necessary because there was no mention of the differences between these three components. A single component A(all) x A(all) could be fitted.

Our estimates of additive variance per trait and cross are not exceptional when compared with those obtained from approaches that have used pedigree or marker-based measures of relatedness for numerous traits in plants, livestock, other model organisms, and humans (e.g. Visscher et al., 2008 PMID 18319743; Yang et al., 2010, among many others). We note that our population of line cross progeny is actually expected to give a higher estimate of epistatic variance when compared to outbred populations: as e.g. Mackay et al., 2014, PMID 24296533 Figure 2 shows, estimates of epistatic variance are maximized as allele frequencies of the interacting loci approach 0.5 (as in our line crosses here). As the reviewer notes, another potential non-additive component, dominance variance, is not accessible in our experiment design which uses haploids, but study designs that can estimate dominance variance have not detected a large contribution (e.g. Parts et al., 2016, PMID 27804950).

We are grateful to the reviewer for pointing out an omission in the Materials and methods section of our manuscript regarding an explanation of why we modeled the epistatic variance with three components. First, as the reviewer suggests, we have added results from a model with a single A(all) x A(all) component to Supplementary file 2. In Author response image 1 we show a visual comparison of the fraction of non-additive variance explained by the three component model (x-axis) and the one component model (y-axis) for each trait and cross (the diagonal line corresponds to identity between the two estimates). The estimates are very similar for most traits and crosses, but one can observe that the three component model occasionally gives a higher estimate. This happens because a key assumption of the one component model – that all pairs of loci contribute to trait variation with effect sizes drawn from a single normal distribution – is violated when one or a few QTL-QTL interactions with large effects are present, resulting in a downward bias. We previously showed (Bloom et al., 2015) that loci involved in such stronger interactions can be detected in additive scans. Therefore, by explicitly including additive QTLs in the three component model, we avoid making the assumption that the effect sizes of all locus pairs are drawn from the same normal distribution and obtain a better estimator of total two-way epistatic variance when large-effect QTL-QTL interactions are present. We have included this rationale in the revised manuscript, in the Materials and methods susbection “Within-cross analysis to estimate additive and pairwise genetic interaction variance”.

3) The two-component mixed model analysis has some caveats. There is correlation between rare and common variants, i.e., the variance components are not orthogonal. This makes any claim about the relative importance of rare versus common less reliable. For example, for the Cadmium chloride trait (Figure 3—figure supplement 1), the 7-component model seems to disagree with the 2-component model, with MAF < 0.01 explaining much less in the 7-component model than in the 2-component model in A. I think comparing the 2-component and one component model could suffer from the same problem. Perhaps a more appropriate (but still not perfect) analysis is to fit single component model first and then fit two component models, do it in two sequential orders (rare, then rare + common; versus common, then common + rare), and look at how the cumulative variance increase. This will tell you which MAF class is more important or can explain more variance. I think in this case, one-component model is more informative than the 2-component model.

We agree with the reviewer that genetic linkage creates a correlation between rare and common variants in genetic mapping studies. The variance component analysis performed here is based on approaches that are the standard in the field, and that have been extensively used in studies of human datasets that seek to address similar fundamental questions about the contribution of variants at different allele frequencies in a population (e.g. Yang et al., 2015; Gazal et al., 2018, PMID 30297966; Wainschtein et al., 2019). How the robustness of estimators obtained from these procedures is affected by the presence of genetic linkage, assumptions about the distributions of causal variant effect sizes, and the relationship between effect size and allele frequency is an active area of research. (e.g. Speed et al., 2017, PMID 28530675) The reviewer is proposing a new forward stepwise variance component analysis which to our knowledge has not been reported before in the literature and which poses its own issues of implementation and interpretation that are beyond the scope of our paper. We agree that this is an interesting idea, and we hope that by making our dataset available, we can stimulate the development of this and other new methods.

With regard to the comparison between the estimates of the contribution of rare alleles from the two-component allele frequency model (light blue bar in Figure 3—figure supplement 1A) and the 7-component model (Figure 3—figure supplement 1B), one can see that for nearly all traits, the estimate of variance explained is very similar, with the exception of cadmium chloride pointed out by the reviewer. We note that cadmium chloride is exceptional among the traits, with nearly all the additive heritability explained by a single locus near the gene *PCA1*, and that the patterns of segregation in different crosses are consistent with allelic heterogeneity at this locus. Contributions of QTLs with large effects are often poorly modeled with whole-genome variance component approaches, and we believe that this accounts for the discrepancy noted by the reviewer.

We further note that the known limitations of variance component analyses were a primary motivation for our study, and that in subsequent sections we also analyzed our dataset using fixed effects models based on detected QTLs. Our study design is highly-powered to detect QTL effects that jointly account for most of the heritable variance, enabling these analyses for the first time. As we show in Figure 3B, Figure 3C, Figure 3—figure supplement 2, and Figure 3—figure supplement 5, the fixed effects models lead to conclusions similar to those obtained from the variance component models.

4) In Figure 3, the relationship between effects and MAF or DAF is a major result. Although many other papers have reported similar results, I think this paper (and the co-submitted paper from Fournier et al.) has the most appropriate design, i.e., the discovery panel is independent of where MAF is estimated. Given its central role in this paper, it probably deserves a bit more clarity. A few questions came to mind when reading this part of the paper. In what analysis is the effect size estimated, single crosses or joint? Could you briefly explain how the effect sizes are estimated in the Results section? If effects estimated from joint analysis, the t-statistic used a factor (n-2) to normalize the degree of freedom, which is smaller for rare variants. This would lead to Beavis effect. I believe the authors used a cross validation strategy to estimate effects, but it’s not very clear by reading the Materials and methods. Can you also plot 2pqa^2 versus MAF? Even if a is large, the variance contributed by rare variants could be small.

We appreciate the reviewer’s positive comments regarding our study design, which decouples allele frequencies in the population from allele frequencies in the mapping panel, thereby allowing us to obtain estimates of effect sizes of rare variants without the typical complications one encounters in GWAS designs regarding sample size and confounding. We welcome the opportunity to clarify the details here, in the revised main text, and in the Materials and methods. Briefly, QTL peak markers are detected in the joint analysis for each trait. Then, for each trait and cross, the phenotypes are scaled to have mean 0 and variance 1, and effect sizes within each cross are estimated using multiple regression for the peak markers that segregate within that cross. The betas in this analysis correspond to the differences in the means between the two QTL alleles. For peak markers that segregate in multiple crosses, the average betas over the different crosses are shown in Figure 3. This is now described in greater detail in the Materials and methods subsection “Effect size estimation for joint QTL mapping”.

The reviewer correctly points out that because we perform model selection and parameter estimation on the same data set, parameter estimates may be upwardly biased (this is known as the Beavis effect). We note that we carried out simulation analyses (Figure 3—figure supplement 2) which indicated that while some estimate inflation is present, it does not qualitatively alter the results in Figure 3. To further address the reviewer’s concern, we have now calculated unbiased estimates of effect sizes by training a model on 9/10 of the data and estimating parameters on the 1/10 of the data held out from the training procedure. The results are shown in a new supplementary figure (Figure 3—figure supplement 4) and are very similar to Figure 3, but estimates are noisier due to the smaller sample size available for unbiased estimation in this procedure. This is now described in the aforementioned Materials and methods subsection.

We believe that it is important to proceed carefully when reporting and interpreting the relationships between allelic effect sizes, variance explained, and allele frequencies for individual QTL effects. We have modified our Introduction to give additional background as to the relevant issues. Figure 3—figure supplement 5 shows the cumulative fraction of variance explained in our mapping panel for each trait by the detected QTLs. In this calculation of variance explained, we used the allele frequency of the QTL peak marker in our mapping population. Were we to instead calculate variance explained in the larger panel of yeast isolates (Peter et al., 2018) using the allele frequencies of variants in that panel, but effect sizes estimated in our mapping population, the variance contributed by rare variants would necessarily be small because the study design severely undersamples variants that are rare and ultra-rare in this larger population (our mapping panel consists of individuals derived from only 16 of 1012 strains). We are concerned that presenting such results would be actively misleading and confusing. We believe that the results from the variance components analysis and those shown in Figure 3—figure supplement 5 would be recapitulated in the larger yeast population if we could detect and estimate the effects of all the variants present in that population, rather than the small fraction that segregates in our crosses.

5) The main conclusion of the manuscript is that rare variants significantly contribute to genetic variance. In my view, this conclusion is biased as these rare causal variants are being analysed in genetic backgrounds in which they are no longer rare; actually, these variants are biallelic. Several studies have shown that a rare variant of MKT1(89A) is a significant contributor to phenotypic variation whenever it is present in segregating populations. However, MKT1(89A) allele hardly identified when one of the parents is not S288c, the strain which harbours this allele. So, the extension that if the rare variant has a significant effect in a sub-population, its effect size would be similar in a large heterogeneous population is false. Furthermore, the authors conclude that in their larger 16 strain segregant populations, a representative distribution of 1000 strain collection, most of the variants have additive effects. This the authors claim is revalidation of their other previous studies (Bloom et al., 2013, 2015), where they identified that most of the causal variants between BYxRM had additive effects. However, in their subsequent paper (Frosberg et al., 2017, PMID 28250458) and another paper (Yadav et al., 2016, PMID 28172852) showed that variance mapping in BYxRM segregants helped to account for genetic interactions and showed how non-additive interactions also contribute significantly to phenotypic variation. Therefore, I find that just doing a few more strains and larger no. of segregants per cross does not make this manuscript a significant advance over the previous studies. One can argue that taking into account all causal variants identified to date (Fay, 2013), one can identify what frequency of rare variants have been identified, e.g. a typical example being MKT1(89A) allele as causal, even though their effect size will not be identified using this strategy. Peltier et al., 2019, show that 284 rare QTNs variants have been identified to date and these functional variants being private to a subpopulation, possibly due to their adaptive role to a specific environment. Moreover, this conclusion can be made without these extensive experimental crosses.

The summary and the previous reviewer comment underscore the novel contributions of our paper, including the ability to address the empirical contribution of rare variants to trait variation in a more comprehensive manner than has previously been possible, and the decoupling of allele frequency from variant discovery. We have taken the opportunity, as requested in the summary, to significantly expand the Introduction to make these and other key points clearer to the non-specialist. We agree that theoretical approaches in evolutionary, population and quantitative trait genetics have been applied to predict the relative contributions of common and rare variants to trait variation under different sets of assumptions; indeed, we cite many relevant papers in our manuscript. We also agree that there is value in aggregating information from individual case reports in the literature – we cited Fay, 2013 in our manuscript, and we noted that the large set of candidate QTGs systematically identified in our study is enriched for QTGs reported in that study. The variants reported in Fay, 2013 represent a sparse sampling of variant effects in yeast, and were by necessity based on studies with small sample sizes, which biased this set of variants in favor of large effects. The review by Peltier et al., 2019 (published after the submission of our manuscript) is similarly based on genes and variants previously reported in the literature. We have added a citation to this paper. We believe that our systematic, comprehensive, empirical approach provides much more general insights into the relative contributions of variants of different frequencies.